# Cataract induction in an arthropod reveals how lens crystallins contribute to the formation of biological glass

**Amartya T. Mitra**[ID]°*, **Shubham Rathore**[ID]¤°, **Augusta Jester, Ruby Hyland-Brown, J. Hassert, Joshua B. Benoit**[ID], **Annette Stowasser, Elke K. Buschbeck**[ID]‡*

Department of Biological Sciences, University of Cincinnati, Cincinnati, Ohio, United States of America

☯ These authors contributed equally to this work.
‡ Lead contact.
¤ Current address: Janelia Research Campus, 19700 Helix Drive, Ashburn, VA, 20147, USA
* mitraat@mail.uc.edu (ATM); elke.buschbek@uc.edu (EKB)

## Abstract

Lenses are vital components of well-functioning eyes and are crafted through the precise arrangement of proteins to achieve transparency and refractive ability. In addition to optical clarity for minimal scatter and absorption, proper placement of the lens within the eye is equally important for the formation of sharp, focused images on the retina. Maintaining these states is challenging due to dynamic and substantial post-embryonic eye and lens growth. Here, we gain insights into required processes through exploring the optical and visual consequences of silencing a key lens constituent in *Thermonectus marmoratus* sunburst diving beetle larvae. Using RNAi, we knocked down Lens3, a widely expressed cuticular lens protein during a period of substantial growth of their camera-type principal eyes. We show that *lens3*RNAi results in the formation of opacities reminiscent of vertebrate lens 'cataracts', causing the projection of blurry and degraded images. Consequences of this are exacerbated in low-light conditions, evidenced by impaired hunting behaviour in this visually guided predator. Notably, lens focal lengths remained unchanged, suggesting that power and overall structure are preserved despite the absence of this major component. Further, we did not detect significant shifts in the *in-vivo* refractive states of cataract-afflicted larvae. This in stark contrast with findings in vertebrates, in which form-deprivation or the attenuation of image contrast, results in the dysregulation of eye growth, causing refractive errors such as myopia. Our results provide insights into arthropod lens construction and align with previous findings which point towards visual input being inconsequential for maintaining correctly focused eyes in this group. Our findings highlight the utility of *T. marmoratus* as a tractable model system to probe the aetiology of lens cataracts and refractive errors.

**Data availability statement:** Manipulation, statistical analysis and graphical representation of data were performed in R v.4.3.2 [77] via RStudio v.2024.04.1 [78]. Code (R and MATLAB) to reproduce all analyses and data figures can be found in the GitHub repository: https://github.com/binturong98/complex-lens-lens3. Raw data are available on the Open Science Framework repository: https://doi.org/10.17605/osf.io/ze7yh [79].

**Funding:** This study was supported by the National Science Foundation, USA under grant IOS-1856241 The funders had no role in study design, data collection and analysis, decision to publish, or preparation of the manuscript.

**Competing interests:** The authors have declared that no competing interests exist.

## Introduction

One of the principal sensory systems essential to the survival of many animals is vision, and this usually involves a dioptric system containing a refractive lens. Owing to their exceptional transparency, lenses have been referred to as 'biological glass' [1]. In vertebrates and invertebrates alike, key constituents of lenses are a collection of densely packed and precisely arranged proteins [2], typically referred to as crystallins. Extensive research has shown that most of these did not first evolve as lens constituents, but rather, were re-engineered from existing components (such as metabolic enzymes, stress-protective proteins and structural components) that vary between taxa to achieve refractive function and optical transparency [3]. This functional convergence explains the considerable diversity in lens proteins across clades, some of which are taxon-specific [4,5]. Regardless of stark compositional differences, lenses are assembled in ways to achieve specific optical properties, with the common objective of producing a sharp image. However, clear vision does not only require transparent lenses, but the coordination of their refractive abilities with retinal placement, so that images of the visual surround are in focus [6,7]. When this is the condition for objects in infinity, the refractive state is referred to as emmetropia. Failure of this coordination leads to refractive errors in which images are focused either in front of the retina (myopia or nearsightedness) or behind it (hyperopia or farsightedness). When uncorrected, refractive errors are often debilitating, causing visual impairment in approximately 101.2 million people and blindness in another 6.8 million [8]. The steady rise of refractive errors, which are linked to other visual deficits such as cataracts and are nearing epidemic proportions [9,10], necessitates further inquiry into refractive state development.

Two mechanisms have been identified by which emmetropia is first established during development, and subsequently maintained (or even refined) as the dimensions and optical properties of the eye change through growth [11]. These are genetic regulation, and active neural feedback in the form of visual input, a combination of which is required to achieve precise optical alignment and minimise image blur [12,13]. Numerous studies which interfere with lens function have shown that the latter is essential in multiple taxa [14,15]. In vertebrates, this has been done by introducing either controlled shifts in focus through the application of dioptric lenses, or image blur using translucent goggles [16]. Altered visual input has also been shown to modulate eye growth in squid, as individuals reared under monochromatic light of different wavelengths develop wavelength-appropriate shifts in eye length [15]. With studies primarily targeting axial length changes, roles of the lens both in focusing and clarity, in establishing emmetropia remain less clear [17]. To provide insights into lens assembly and its roles in refractive development, we focused on the structurally somewhat simpler and relatively understudied corneal lenses of an arthropod. Unlike vertebrate lenses which can typically accommodate to change focus and are composed of different cell types (such as lens fibre and cuboidal cells) [18], most arthropod lenses are cuticular and consist of chitin fibrils that are arranged in concentric paraboloids within a protein matrix [19,20]. The outer surfaces of these miniature lenses are continuous with the head cuticle [19].

Arthropod lens development and maintenance is best known from the compound eye lenses of *Drosophila melanogaster*, in which corneal lens proteins are secreted from microvillar projections on the apical surfaces of support cells (Semper or cone cells and primary pigment cells) [21,22]. Studies on *D. melanogaster* and other arthropods have shown that these cuticular lens proteins likely vary in abundance in different regions of the lens, contributing to its precise construction [23–25]. A noteworthy example is found in the unique camera-type principal eyes of *T. marmoratus* larvae [26] (Fig 1A, B), the only extant species shown to possess a bifocal lens [27]. This complex lens is secreted by support cells (eye glia) which form the tubular, vitreous-like portion of the principal eyes [28,29], at the proximal end of which lies the retina (Fig 1C). Based on mass spectrometry the lens is composed of ten main proteins [30], the majority of which are cuticular proteins, a large group of proteins that also play roles in other insect lenses such as those of *Anopheles gambiae*, *D. melanogaster* and *Tribolium castaneum* [23,24,31]. In *T. marmoratus*, specific lens proteins are expressed to varying degrees in different regions of these support cells [30].

Here, we explore the role of a key lens protein, Lens3 (Tm-lens3 or TmarCPR1) and specifically, its role in post-embryonic lens development and establishment of correct focusing in *T. marmoratus* principal eyes. Lens3 is a chitin-binding protein of the RR-2 subfamily, an expansive class of proteins [32] that includes 3 out of 4 of the *D. melanogaster* lens proteins [24], and hence may be a particularly important component of arthropod lenses. Proteomics has shown that, in addition to forming heteromeric complexes with other lens proteins, Lens3 is the most abundant and widely expressed of the different proteins constituting the larval lens, and therefore, is likely one of its integral components [30]. Accordingly, we investigated the potential consequences of RNAi-mediated *lens3* knockdown in *T. marmoratus* during the 2nd to 3rd

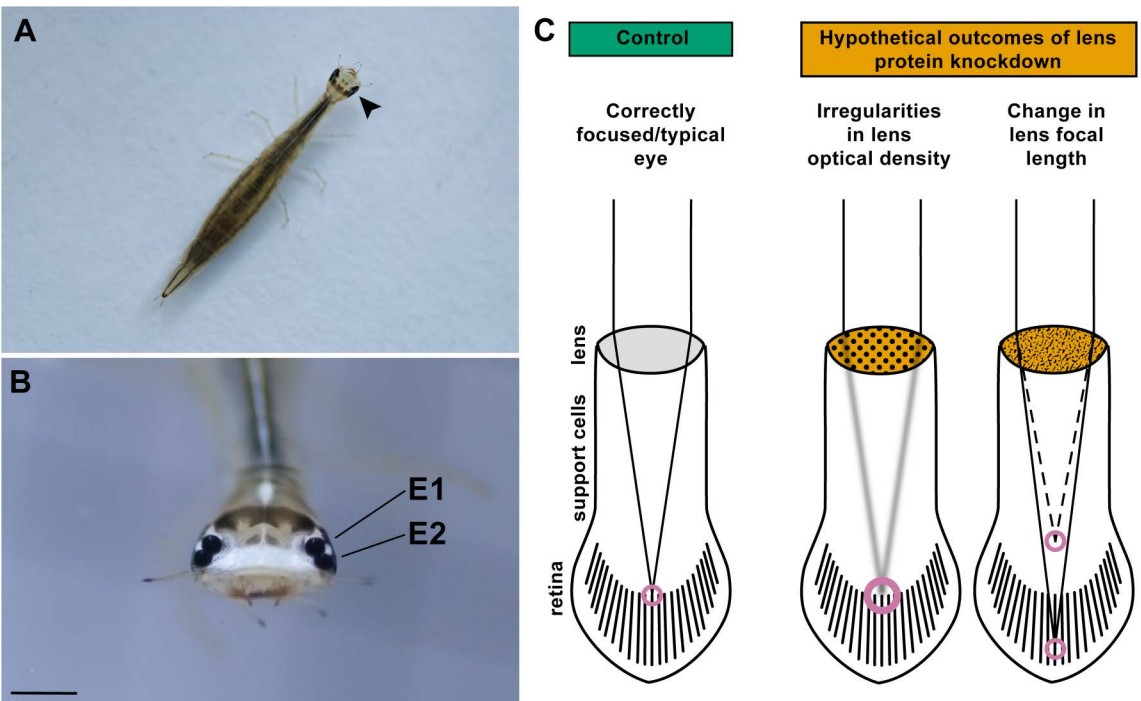

**Fig 1. Principal eyes in *T. marmoratus* larvae and possible outcomes of reduced levels of a widely expressed lens protein during postembryonic growth.** A) Third instar *T. marmoratus* larva. Arrowhead points to the principal eyes, eye 1 (E1) and eye 2 (E2) which are enlarged in (B). Scale bar = 1 mm. (C) Schematics of larval eyes in which light projected by the lens may fall differently onto the retina, depending on possible outcomes of silencing *lens3*. Eyes of control individuals are expected to be focused near the top of the retina. However, disruption of lens structure through *lens3*R-NAi may cause 1) irregularities in lens optical density leading to the projection of degraded images onto the retina or 2) hyperopia (farsightedness; solid line) or myopia (nearsightedness; dashed line) due to altered refractive power of the lens.

instar developmental transition, which is when the lengths of the principal eyes increase rapidly by ~ 30–35%, and the lenses undergo coordinated growth [33]. Chiefly, we examined if the loss of this major lens protein could lead to either a shift in focus or blurring (Fig 1C), adopting an integrative approach with the examination of lens morphology, lens optics, *in-vivo* refractive states of the eyes, and hunting behaviour.

## Results

### Knockdown of a key lens protein gene induces the formation of 'cataracts'

For molecular confirmation of knockdown, we used qPCR, and found that RNAi via dsRNA injection (both probes; Fig 2A) lead to a significant reduction in *lens3* mRNA transcripts compared to controls, which were injected with buffer solution (Fig 2B; ANOVA: $F_{2,12} = 252.46$, $p < 0.001$). Larval viability was unaffected by *lens3*RNAi (mortality; $n_{control} = 2/42$, $n_{lens3RNAi} = 2/69$) but physical examination of principal eyes and densitometry of dissected lenses revealed the presence of opacities in the posterior lens surfaces which were localised to the centre in 100% of treatment individuals (See Fig 3C, S1 in Supplementary Information S1 File). Based on appearance (see below) this phenotype is reminiscent of vertebrate lens opacification [34]. Notably, opacities are most similar to human posterior subcapsular cataracts owing to their location and spread [35–37]. We therefore refer to these deficits, which were exclusively observed in treatment individuals, as 'cataracts'.

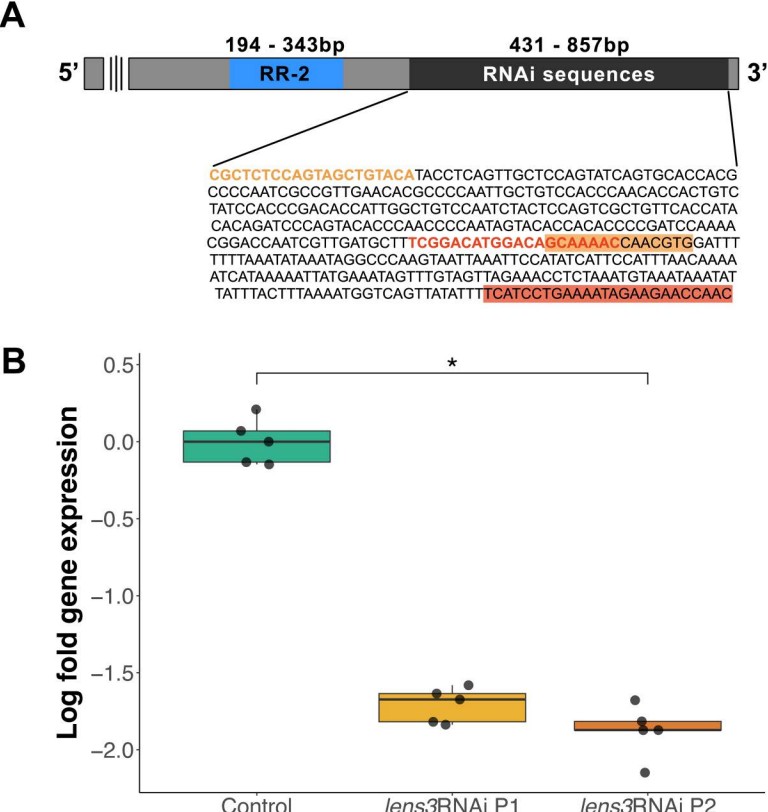

**Fig 2. T. marmoratus *lens3* gene and RNAi mediated knockdown.** A) *lens3* codes for a cuticular chitin-binding protein in the RR-2 subfamily characterised by the conserved Riebers and Riddiford consensus sequence [30] (depicted in blue). Sequences used to generate amplicons for RNAi are located towards the 3' end. Primer binding regions for probe 1 (*lens3*RNAi P1) are highlighted in orange and red, while those for probe 2 (*lens3*RNAi P2) are depicted using orange and red text. B) Validation of RNAi induced knockdown using quantitative PCR (qPCR). Log fold gene expression is significantly lower in individuals injected with *lens3* dsRNA (orange and red) compared with control individuals injected with buffer (green; $n = 5$ individuals per group, $p < 0.001$).

## Cataracts cause severe image degradation but do not impact lens focal length

To determine the effects of opacification on focusing properties of lenses, we performed a series of optical assessments. Specifically, we evaluated focal distance and the quality of images projected by lenses using a modified version of the hanging drop technique [27]. A series of photographs were captured from the back surface of lenses to the images produced by them at regular 5μm intervals (Fig 3A). By computing grayscale values across these image series [27], we found that lenses with cataracts projected degraded and blurry images (Fig 3B–E) with lower edge sharpness (Fig 3F, G; Kruskal-Wallis: $\chi^2_{LE2} = 19.8$, d.f. = 2, $p_{LE2} < 0.01$). However, measurements of back focal length, or the distance from the back surface of lenses at which best focused images are formed (indicated by highest edge sharpness values and verified with

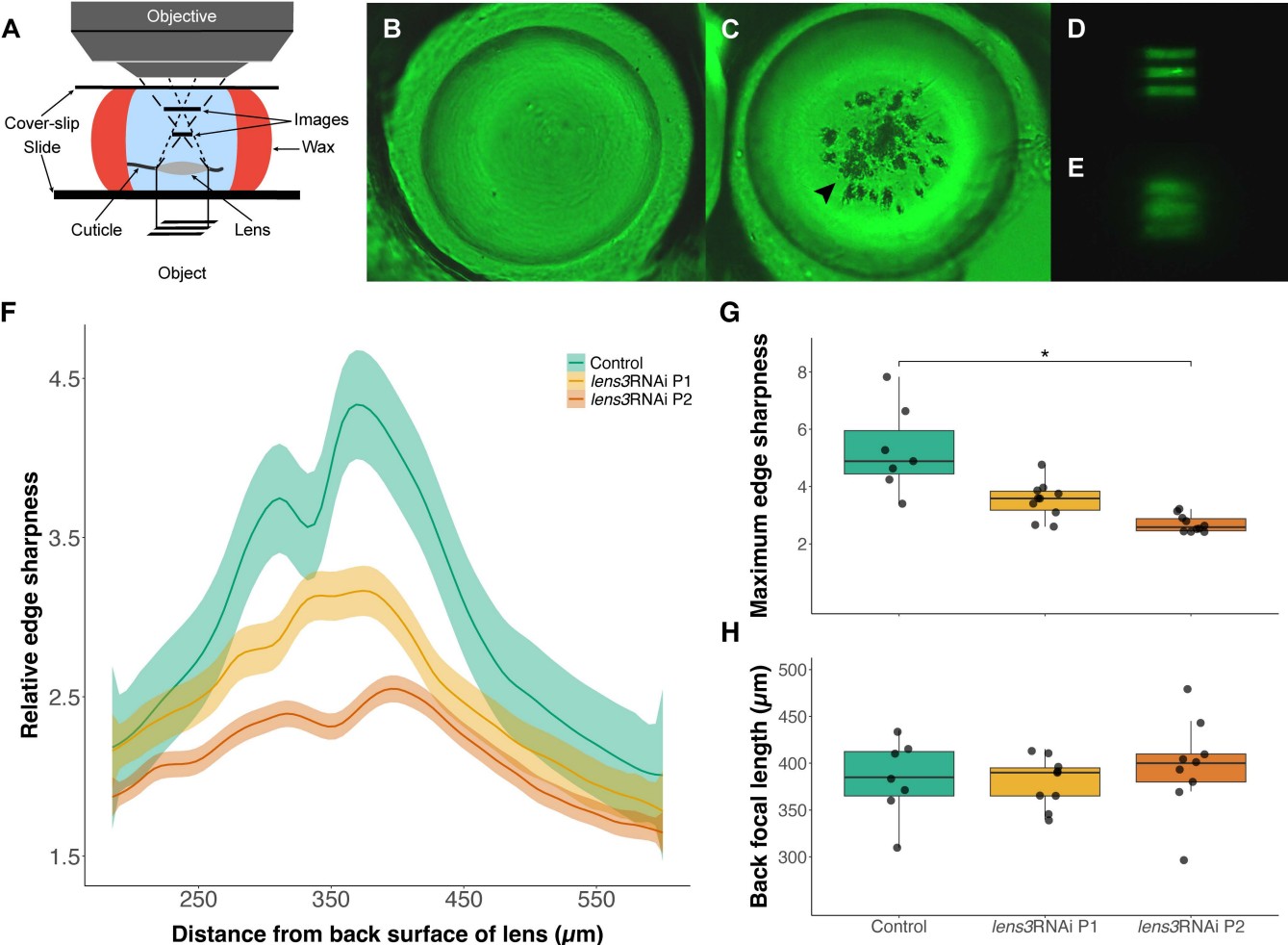

**Fig 3. Assessment of lens optics following cataract induction.** A) Schematic of a modified version of the hanging-drop method [27] to obtain a series of photographs from the back surface of dissected lenses to the projected images of a grating. When mounted this way, lenses of control individuals are clear (B; $n_{control} = 40$) and project sharp images (D), while those of *lens3*RNAi individuals have visible defects near the centre (C; arrowhead; $n_{lens3RNAi} = 67$) and produce degraded and blurry images (E). Measurements of relative edge sharpness (see methods for details) corroborate this finding of image blur, as depicted by weighted average (locally estimated scatterplot smoothing or LOESS) curves (F; $n_{control} = 9$, $n_{lens3RNAi\ P1} = 10$, $n_{lens3RNAi\ P2} = 11$). Shaded areas represent standard error. Maximum edge sharpness values which correspond to the highest peak values from individual curves showed significance (G; $n_{control} = 9$, $n_{lens3RNAi\ P1} = 10$, $n_{lens3RNAi\ P2} = 11$, $p_{LE2} < 0.01$). H) In contrast, the distance from the back surface of the lens at which best focused images are formed (also obtained from peak edge sharpness values in F) are not significantly different across groups (G; $n_{control} = 9$, $n_{lens3RNAi\ P1} = 8$, $n_{lens3RNAi\ P2} = 9$, $p = 0.62$) indicating no detectable shifts in focal length.

visual assessment) were not significantly different (Fig 3H; ANOVA: $F_{LE2; 2,24}=0.47$, $p_{LE2}=0.62$). This indicates that *lens3*R-NAi does not cause detectable shifts in focal length. Additionally, the bifocal nature of the lens was retained with *lens3*R-NAi, evidenced by the presence of two peaks for relative edge sharpness in Fig 3F. For ease of representation, the data presented here are restricted to LE2, with data for other eyes (LE1, RE1, RE2) presented in Fig S2 in S1 File.

## Ultrastructural characterisation of cataracts

For examination of cataracts at the ultrastructural level, we used electron microscopy (EM). The exterior regions of dissected lenses were examined using scanning EM (SEM) which revealed abnormalities at the proximal surfaces or undersides of lenses. While lens surfaces of control individuals were smooth, those of *lens3*RNAi individuals were characterised by rough and uneven surfaces, ridges, spokes, perforations, and in severe phenotypes, sunken surfaces (Fig 4). In addition, to determine the extent to which defects penetrate into the lens, we obtained transmission EM (TEM) micrographs of sagittal eye sections (Fig 5A). Micrographs of control lenses appear typical to those of arthropods, which are cuticular in nature [22,38]. The lens is characterised by alternating lamellae of chitin microfibrils (lightly stained) embedded in a helicoid arrangement within a secreted protein matrix [19] (darkly stained; Fig 5B, D, G). In contrast, sections of *lens3*RNAi individuals revealed widespread disorganisation in this lamellar construction, with defects extending from lens surfaces into the interior (Fig 5C, E, F, H, I). Numerous large, darkly stained defects are visible towards the proximal surface and centre, which appear to disrupt lens microarchitecture (Fig 5; arrowheads and asterisks). No such disturbances were visible in the periphery (Fig 4, S3 in S1 File) based on SEM and TEM observations.

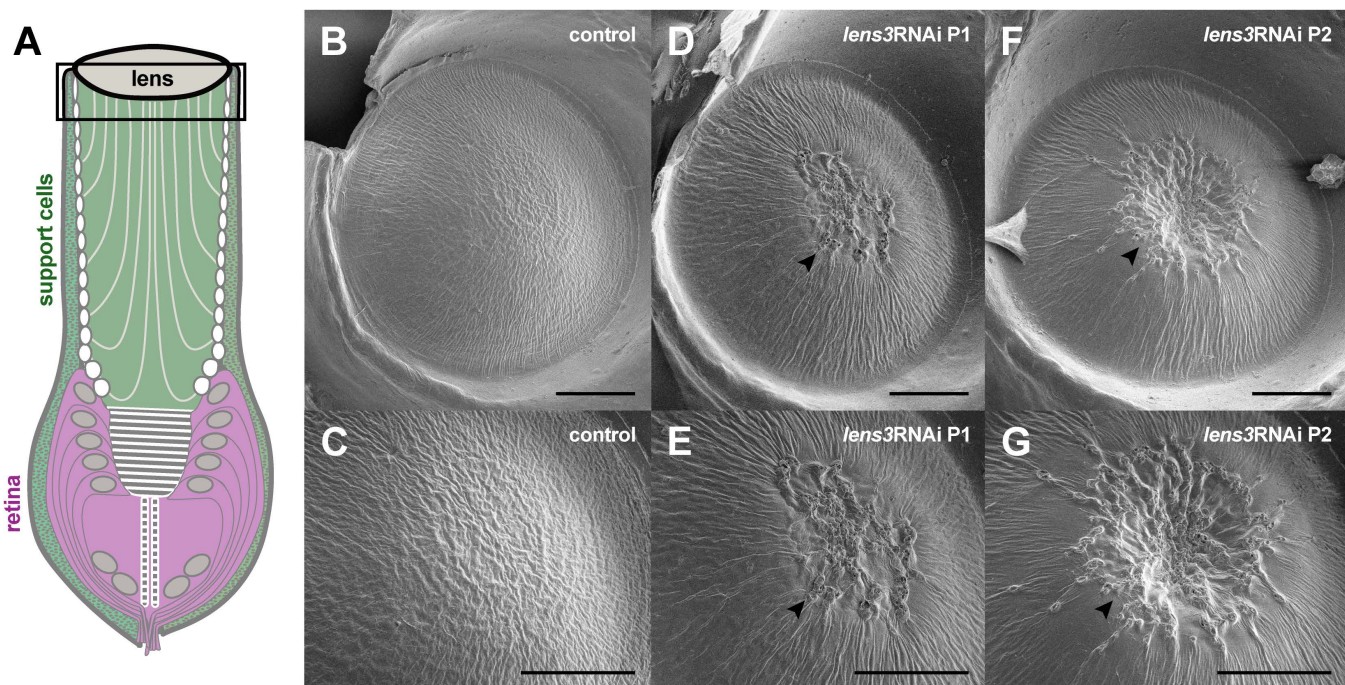

**Fig 4. *lens3*RNAi causes centrally localised 'cataracts' on the proximal surface.** A) Schematic indicating the location of SEMs depicting the underside of the lens. SEM images of the surfaces of dissected and fixed lenses are smooth and regular in control individuals (B,C; $n_{control}=5$). Lens defects are observed in 100% of treatment individuals (D-G; $n_{lens3RNAi\,P1}=5$, $n_{lens3RNAi\,P2}=5$). Lens undersides of *lens3*RNAi individuals are irregular, perforated (arrowheads) and in severe phenotypes, appear sunken (arrowheads in F,G). Each column represents the same lens photographed at different magnifications (scale bars=50μm).

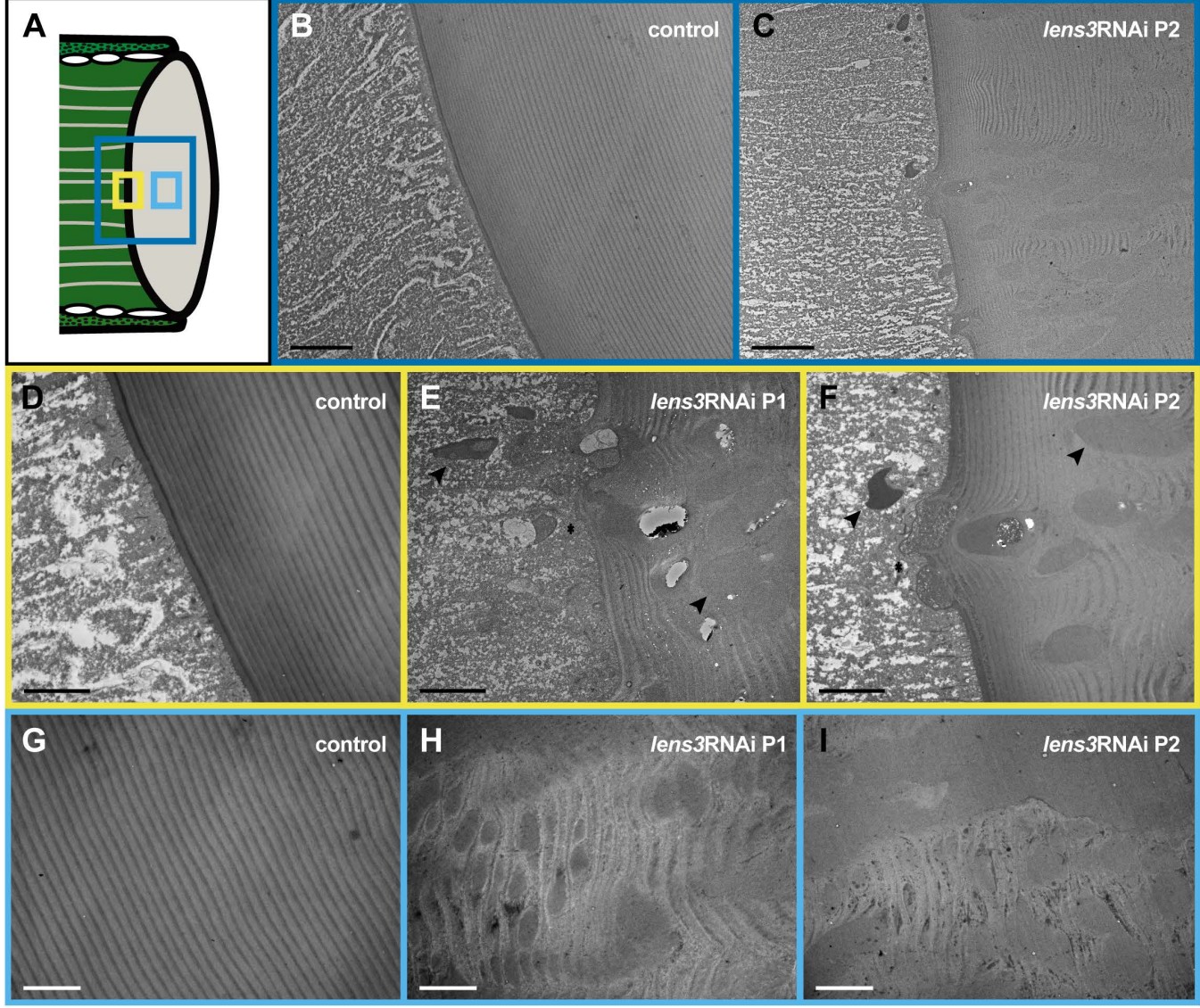

**Fig 5. Widespread disorganisation of lens architecture caused by *lens3*RNAi.** A) Schematic diagram with frames depicting regions of principal eye sagittal sections captured in transmission electron micrographs. Dark blue, yellow and light blue frames correspond to overview (B,C), proximal surface of lens (D,E,F) and lens interior (G,H,I) regions respectively. The lenses of control individuals (B,D,G; $n_{control}$ = 3) are characterised by alternating lamellae of chitin microfibrils (lightly stained) embedded in an electron dense protein matrix (darkly stained), illustrating the helicoid architecture that is typical of arthropod cuticular lenses [19]. In treatment individuals (E,F; $n_{lens3RNAi\ P1}$ = 2, $n_{lens3RNAi\ P2}$ = 2), this precise lamellar arrangement is disrupted by large, darkly stained areas (likely protein aggregates; arrowheads) which appear to fail to be appropriately incorporated into the lens (asterisks) and extend into the lens interior (H,I). Scale bars = 10μm in B,C; 4μm in D-I.

## Refractive state of the eye is not altered by cataractogenesis

Using a custom-built micro-ophthalmoscope [39], we then assessed whether cataract induction affected the refractive state of principal eyes [39]. Exploiting the autofluorescent properties of arthropod photoreceptors, with this technique we visualised the linear proximal retina of live larvae to measure the distance at which objects appear best in focus. Although the proximal retinae of treated larvae lead to blurry images compared to controls (Fig 6A, B), we found that they had

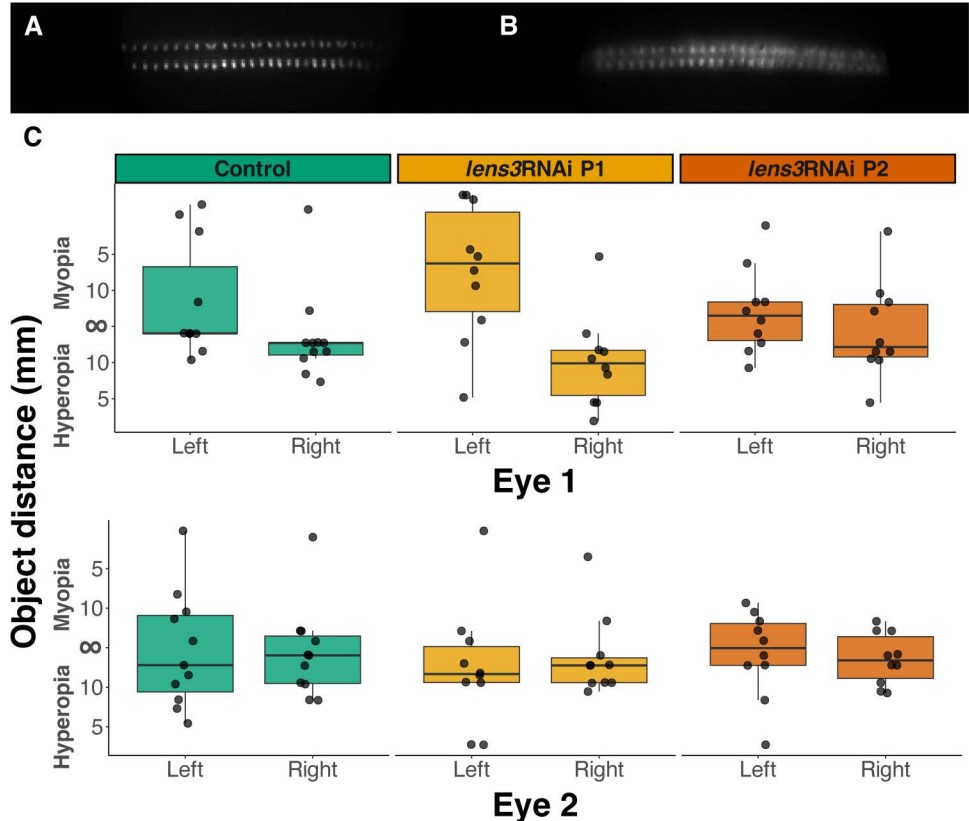

**Fig 6. Cataract induction does not affect the refractive state of principal eyes.** Ophthalmoscope images of the proximal retinae of control individuals (A) are sharp whereas those of *lens3*RNAi individuals (B; *lens3*RNAi P2 depicted here) are blurry in comparison. However, there are no significant differences between the best focused object distances of the principal eyes of control and *lens3*RNAi individuals, assessed by visualising the proximal retina using micro-ophthalmoscopy [39] (C; $n_{control} = 11$, $n_{lens3RNAi\ P1} = 10$, $n_{lens3RNAi\ P2} = 11$; $p_{LE1} = 0.21$, $p_{LE2} = 0.56$, $p_{RE1} = 0.23$, $p_{RE2} = 0.93$).

similar refractive states (Fig 6C; Kruskal-Wallis: $\chi^2_{LE1} = 3.05$, d.f. = 2, $p_{LE1} = 0.21$; $\chi^2_{LE2} = 1.13$, d.f. = 2, $p_{LE2} = 0.56$; $\chi^2_{RE1} = 2.89$, d.f. = 2, $p_{RE1} = 0.23$; $\chi^2_{RE2} = 0.12$, d.f. = 2 $p_{RE2} = 0.93$).

## Hunting behaviour of cataract afflicted larvae is only impacted in dim light environments

As *T. marmoratus* larvae are visually guided predators, we evaluated their hunting behaviour to determine if *lens3*RNAi, with the resulting cataracts and image blur, is of functional significance. For this, as in Rathore et al., 2024 [40] we presented individuals with mosquito larvae (prey) and recorded video footage of their hunting. During trials conducted under broad spectrum bright lighting conditions (Fig S4 in S1 File; see methods for details), our treatment did not affect hunting behaviour. This included quantification of latency (the time it took to hunt the first prey item; Fig 7A, C; Wilcoxon rank-sum; $z_{horizontal} = 89.5$, $p_{horizontal} = 0.52$; $z_{vertical} = 122$, $p_{vertical} = 0.98$). Hunting success was also similar between control and test individuals for both arenas (Fig 7E, G; Wilcoxon rank-sum; $z_{horizontal} = 109$, $p_{horizontal} = 0.28$; $z_{vertical} = 157$, $p_{vertical} = 0.23$). However, when larvae were challenged by being made to hunt in dimmer light environments, we found a significant increase in latency for treated individuals (Fig 7B, D; Wilcoxon rank-sum; $z_{horizontal} = 77$, $p_{horizontal} = 0.03$; $z_{vertical} = 44$, $p_{vertical} = 0.009$). Further, we observed a weak (nonsignificant) decrease in the hunting success of treated larvae (Fig 7F, H; Wilcoxon rank-sum; $z_{horizontal} = 154$, $p_{horizontal} = 0.22$; $z_{vertical} = 128$, $p_{vertical} = 0.42$). See Fig S4 in S1 File for spectral composition of dim and bright light environments in both arenas.

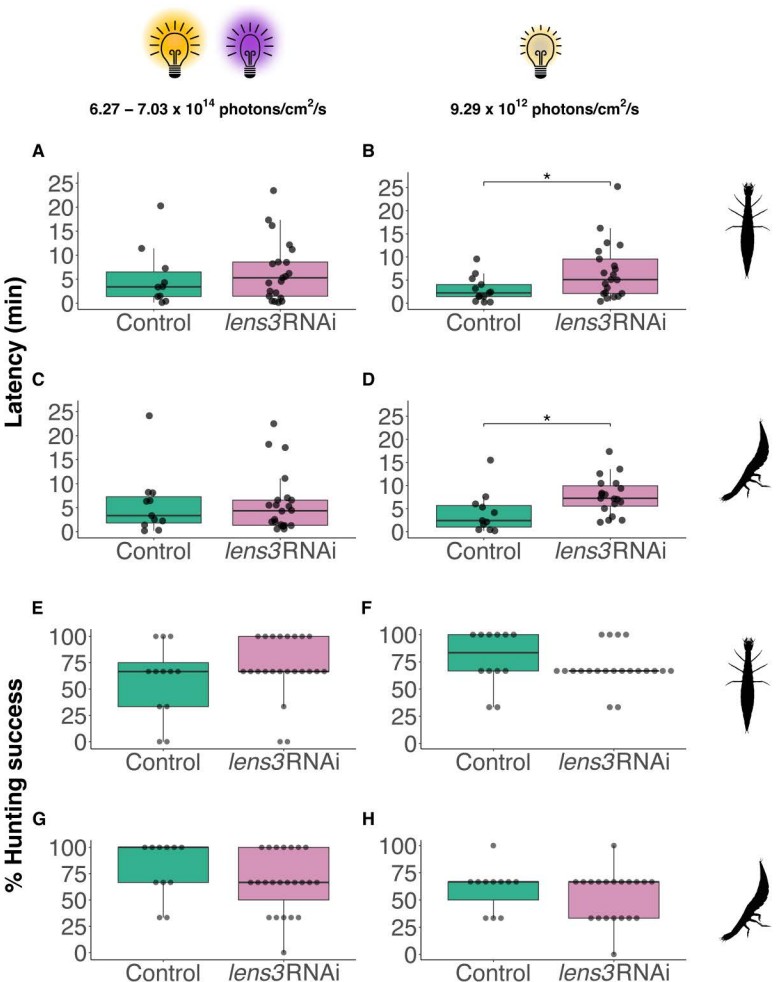

**Fig 7. Cataract induction does not affect hunting behaviour in optimal light conditions (left panels) but causes impairments in challenging light environments (right panels).** Latency, defined here as the time taken for the first successful hunt when presented with prey items, is comparable between control and treatment individuals in horizontal (A; $n_{control}$ = 12, $n_{lens3RNAi}$ = 23, $p_{horizontal}$ = 0.52) and vertical (B; $p_{vertical}$ = 0.98) setups under bright lighting conditions with UV supplementation. However, latency is significantly higher in *lens3* RNAi individuals under dim white light in both arenas (B,D; $n_{control}$ = 14, $n_{lens3RNAi}$ = 22, $p_{horizontal}$ = 0.03, $p_{vertical}$ = 0.009). Similarly, under bright and UV light, we did not detect differences in hunting success (E,G; $n_{control}$ = 12, $n_{lens3RNAi}$ = 23, $p_{horizontal}$ = 0.28, $p_{vertical}$ = 0.23), but observed a weak (nonsignificant) decrease in the hunting success of treatment individuals under dim white light (F,H; $n_{control}$ = 13, $n_{lens3RNAi}$ = 22, $p_{horizontal}$ = 0.22, $p_{vertical}$ = 0.42).

## Discussion

### The integral role of lens proteins in maintaining lens transparency

Lenses are precisely constructed transparent structures which improve the resolution and sensitivity of many animal eyes through their unique refractive abilities. Amongst invertebrates, the molecular constitution of lenses is highly variable, ranging from crystallin proteins such as those of cephalopods and cubozoans (jellyfish) [41] to mineral aragonite in chiton molluscs [42]. In this relatively understudied field, work on the makeup of arthropod lenses has focused primarily on insects (with the exception of a single study on copepods [43]). Species examined include *D. melanogaster* [24], the mosquito *A. gambiae* [25], the beetle *T. castaneum* [23], and *T. marmoratus*, the focal species of this study [30]. In all these cases, the lenses contain highly variable numbers and classes of cuticular lens proteins, some of which have

ancient origin and have been shown to serve other functions [23,44]. Considering this diversity, it is unclear how animal lenses converge onto the common function of minimising light scatter and building what has been termed as 'biological glass' [1]. Valuable morphological, molecular and developmental insights stem mostly from studies on vertebrates in which the genes encoding for specific lens components were silenced [45–47]. Although, beyond measurements of how such treatments affect refractive index [48,49], consequences for visual function remain understudied. Studies exploring the contribution of individual lens components to overall form and function in more taxa are thus required for a deeper understanding of this question. Here, we observed that RNAi knockdown of *lens3* in *T. marmoratus* larvae during the second to third larval instar transition, (a period of substantial eye and lens growth [33]), resulted in the formation of cataracts without exception. Defects were localised to the proximal region of the lens, where the bulk of lens secretion occurs during post-embryonic growth [21,28]. Notably, despite widespread expression of *lens3* [30] throughout principal eye support cells (situated below the entire lens surface), the most severe knockdown effects were in the central portion of the lens (Fig 4, and S1 in Supplementary Information SI in S1 File). This suggests that the most substantial deposition of new material is primarily in this region or that other processes are in effect that keep the periphery of the lens intact (See Fig S1, S3 in Supplementary Information SI in S1 File). Through examination of ultrastructure, we speculate that deformities are likely due to the formation of large aggregates which fail to be adequately incorporated into the lens, thereby deteriorating its distinct lamellar construction. Notably, no defects were found near the outer (distal) lens surface. Here, the portion of the lens that is continuous with head cuticle is shed during moulting [19] and pre-existing underlying layers likely form the new surface post-moult (Fig S3 in S1 File). We show that disruption of protein and chitin organisation increases heterogeneity within the lens and manifests in the form of pockets of opacities rather than gross changes in refractive index or the overall radius of curvature. Presumably this preserves the integrity of large portions of the lens, which then retains normal refractive power. Localised lens defects lead to scattering of light, which, as in vertebrate cataracts, caused image degradation. This is evidenced by the projection of blurry and degraded images as opposed to changes in overall focal length (Figs 1C and 3F–H). Lens3 thus plays an important role in maintaining lens transparency, although determining whether deficiencies result from protein misfolding or anomalous interaction of multiple lens components [50] requires further *in-vitro* studies. Considering the pleiotropy and dynamic expression patterns of cuticular and lens protein genes [51], precise functions of Lens3 may be also identified by determining its localisation within the lens and the effects of its knockdown at earlier stages such as during embryonic development. These experiments might also provide insights into the bifocal nature of the larval lens, which was curiously retained with *lens3*RNAi. Investigations involving the suppression of other lens protein genes may shed light onto how this unique optical structure is crafted.

Although the lenses of both arthropods and vertebrates are transparent, numerous and stark differences exist between them. Arthropod lenses are secreted and consist of chitin fibrils within a matrix of cuticular proteins. In contrast, vertebrate lenses are cellular and contain high concentrations of soluble crystallins with various non-lenticular enzymatic and chaperone functions. Despite this, our results bear resemblance to those of crystallin mutants and knockouts in mice and zebrafish [52–55]. Most notable here is that loss of function mutants of α-crystallins, one of the highly expressed proteins in vertebrate lenses, also develop centrally localised protein aggregate cataracts, suggesting that similar principles of macromolecular organisation may be involved to achieve lens properties.

### Consequences of cataract induction on eye optics and vision

To examine the importance of visual feedback to establish and maintain emmetropia, multiple studies have evaluated the effects of altered visual input during growth. One way this is done is by imposing form deprivation, i.e., preventing the retina from receiving clear images during growth. Typically this is accomplished through the application of translucent goggles or frosted diffusers. Although the diversity of species (fish, birds and mammals) in which this has been examined is impressive (for review see Troilo et al., 2019 [16]), until now, vertebrates remain the sole clade subjected to these treatments. In these studies, experimental animals generally tend to develop axial myopia when their retinae are presented

with degraded images. This approach, however, is particularly challenging in arthropods due to their small eye size and intermittent rapid growth periods which involve shedding of the entire exoskeleton (including surface eye cuticle). Here we show that molecular manipulation of the lens caused dramatic levels of form deprivation in *T. marmoratus* and our measurements indicate that the image blur from *lens3* suppression induced cataracts is substantial, even impacting larval behaviour. Although hunting behaviour appeared unchanged in initial experiments, these were conducted under conditions amenable to larvae, in brightly-lit environments well-suited for their visually guided predation. However clear deficits in latency were detected when the same trials were conducted in dimmer, somewhat challenging conditions. These findings align with the decreased contrast sensitivity and low contrast visual acuity associated with cataracts, which are known to be exacerbated under low luminance [34,56,57].

Despite substantial deficits, we found that induced form deprivation did not lead to significant shifts in the refractive states of larvae. Although in contrast to findings in vertebrates [12], this aligns with previous work in which rearing this species and other arthropods in darkness (which denied them visual input) did not affect refractive state [58]. Considering that back focal lengths remained unchanged (Fig 3H), compensatory changes in eye axial length are unlikely. Instead, our findings provide additional support to the hypothesis that the development of correctly focused arthropod eyes does not rely on visual feedback but is instead primarily regulated by genetic processes [58]. Without the confounding process of active visual input, arthropods are thus particularly well-suited models to probe the physiological processes and underlying molecular mechanisms involved in establishing and maintaining emmetropia. For instance, a recent study on *T. marmoratus* larvae showed that osmotic processes in the support cells play a key role in proper development of the axial length of their principal eyes [40]. As intraocular pressure has long been implicated to drive postembryonic eye growth in vertebrates [59], this mechanism may be of ubiquitous importance for proper refractive development [60]. These findings are substantiated by a recent transcriptomic analysis that also revealed the expression of osmoregulatory genes within the larval principal eyes [29]. However, functional studies of these genes are needed to further dissect possible mechanisms. While multiple screening and expression efforts in traditional vertebrate models have revealed clues into the genes and loci regulating proper eye growth, a broader framework underlying cellular pathways and signalling cascades remain unclear [61–63]. The need for such a framework is fueled by studies that have already shown strong correlations between the transcriptional regulation and signalling pathways that are involved in eye and lens morphogenesis of both vertebrates and invertebrates [21]. This is corroborated by a growing consensus that specific aspects of eye development are deeply conserved across animals [64–66].

## Conclusions

Successful vision is reliant on highly transparent and optically dense lenses which are formed by a variety of tightly packed and intricately arranged proteins. While much emphasis is placed on understanding how defects in lens composition lead to vertebrate eye cataracts, our study is amongst the first to illustrate that similar defects are inducible in invertebrates. Knockdown of a major lens protein caused localised structural defects, leading to image degradation and behavioural deficits in low-light conditions as observed in vertebrates. Unlike vertebrates, form deprivation, or image blur during growth did not result in long-term adjustments to the refractive state of the eye, corroborating previous findings that visual feedback does not contribute to the establishment of correct focus in arthropods [58]. Our work adds to the body of literature exploring deep conservation of the salient features of eyes [11,67–69] by highlighting both convergence in the mechanisms of lens assembly and function, and divergence in the processes regulating coordinated eye growth.

## Methods

### Insect husbandry and RNAi

Experimental *T. marmoratus* larvae were isolated from a lab-reared colony at the first instar stage and housed in separate enclosures due to their cannibalistic nature. Individuals were maintained at 25–27°C with a 14:10 hour

light-dark cycle and reared on a diet of frozen bloodworms, brine shrimp and live mosquito larvae (*Aedes aegypti*). For RNAi, two dsRNA probes against *lens3* (to account for possible off-target effects) were generated from a published mRNA transcript [30]. 191 bp (*lens3*RNAi P1) and 249 bp (*lens3*RNAi P2) long unique regions (outside of the RR domain) were identified using NCBI BLASTx [70] and subsequently amplified (see Table S1 in Supplementary Information SI in S1 File, Fig 2A for gene and primer sequences) using whole-head cDNA. These amplicons were used for dsRNA synthesis as detailed in Rathore et al., 2020 [71]. To ensure knockdown of *lens3* during eye and lens growth at the time of moulting into the third instar, 4 μg of *lens3* dsRNA was injected in one-day old second instar larvae. All data were collected from third instar larvae one day post-moult. Control individuals were injected with buffer solution [71].

### RNA isolation and confirmation of *lens3*RNAi

For RNA extraction, individual heads were dissected in RNA*later*™ (Invitrogen) and stored in TRIzol™ reagent (Ambion) in −20 °C until further processing. Total RNA was isolated using the RNeasy Lipid Tissue Mini kit (Quiagen) and treated with DNAse to exclude the possibility of DNA contamination. It then was used to synthesise cDNA using the Omniscript RT Kit (Quiagen) in accordance with manufacturer instructions. For knockdown validation, quantitative PCR (qPCR) assays were conducted on 300 ng of prepared cDNA using the Maxima SYBRGreen/ROX kit (Invitrogen). Reactions were carried out in triplicates per template in a final volume of 20μl on the QuantStudio™ 3 Real-Time PCR System (Applied Biosystems). The following cycling parameters were used: one cycle at 95°C, 40 cycles of denaturation at 95°C, annealing and extension at 60°C. Relative expression was normalised to the *RpL13A* reference gene. Sequences for gene-specific primers used are available as table in Supplementary Information S1 S1 File. Differences in *lens3* transcript levels between control and treatment groups were tested using a one-way ANOVA.

### Image contrast, back focal length and quantification of cataract

Optical assessments of lenses were carried out using a version of the hanging drop method modified in the lab [72]. Briefly, excised lenses were mounted with wax and submerged in a 50% dilution of insect Ringer solution (to preserve their integrity [27]) between a glass slide and coverslip (Fig 3A). This arrangement was then placed on a microscope stage from which the condenser was removed and illuminated with monochromatic light. First, a 1x1 cm square wave grating (USAF 1951 negative test target from Edmund Optics, Barrington, NJ, USA) as an object was placed 12.2 cm away from the lenses. A high-resolution camera (Moticam 3.0) and time-lapse software (Motic Images Plus 2.0.25) were then used to capture a series of photographs, at 5 μm intervals, from the back surface of lenses to images projected by the lenses. To quantify the severity and location of cataracts, we used digital evaluation of lens back surface images with densitometry tracing, a method for cataract quantification adapted from Seeberger et al., 2004 [73]. Briefly, using the image processing software Fiji [74], a single horizontal line of pixels was selected through the lens and the plot profile function was applied to generate pixel brightness or grey values along all points on the selected lines (See FigS1A-C in Supplementary Information SI). Grey values of centre and surround points were extracted for groupwise comparison and analysed using one-way ANOVA tests (See FigS1D,E in Supplementary Information SI in S1 File). Image series were then processed using a custom-made MATLAB script (Matlab 2023, The Mathworks, Natick, MA) to determine relative edge sharpness (detailed in Stowasser et al., 2010 [27]). Briefly, this method computes the grayscale intensity values of pixels across the image stack to evaluate which photographs in the series have the highest edge sharpness (corresponding to sharp, best focused images projected by lenses). This was then used to determine back-focal distances of the second, sharper image produced by the bifocal lens, which typically is focused on the proximal retina [27]. These were analysed using one-way ANOVA tests. Kruskal-Wallis tests were used to compare maximum sharpness values between groups. Image series with improper positioning of lenses were excluded from analysis.

## Electron microscopy (EM)

To assess lens ultrastructure, we conducted EM. Lens surfaces were examined using SEM, and individuals were anaesthetised on ice prior to dissecting principal eye lenses and surrounding cuticle from the head capsule. To prevent cracking from desiccation, lenses were first fixed in 4% paraformaldehyde and 3.5% glutaraldehyde solution in 0.2 mol L-1 Sorensen's buffer for 16–18 h at 4°C. After washing thoroughly with distilled water, fixed lenses were dried at room temperature for 24 h and mounted to expose the inner surfaces (undersides) on stubs with adhesive carbon pads. Samples were sputter coated with gold and imaged using a scanning electron microscope (FEI Apreo LV-SEM). To examine the interiors of lenses, we performed TEM, for which larval heads were dissected, fixed, and prepared for sectioning using standard protocols described in Stowasser and Buschbeck, 2012 [75,76]. Images were acquired using a transmission electron microscope (Hitachi H-7650). Brightness and contrast on EM images were adjusted using Adobe Photoshop 2022.

## Ophthalmoscope imaging

To assess the refractive states of larvae, we used a custom-built micro-ophthalmoscope based on the inherent autofluorescence of arthropod photoreceptors [39]. In brief, animals were positioned with principal eyes (E1, E2) looking into a UMPlanFL 10x water immersion objective, with the lens surface in focus of the camera. Then, an accessory lens was added to visualise the linear proximal retina (red-autofluorescent photoreceptors) when illuminated through a Texas Red filter. Image series of the retina were obtained at regular intervals while moving the accessory lens along a rail, and the position at which a focused image of the retina was obtained, determined the refractive state of the eye [39]. The relationship between accessory lens positions and corresponding in-focus object distances was obtained from a calibration curve established for the objective. All four principal eyes were imaged, and best-focused images were chosen blindly to eliminate bias. Kruskal-Wallis tests were used for statistical comparison of object distances.

## Assessment of hunting behaviour

We analysed *T. marmoratus* hunting behaviour using methods adapted from Rathore et al., 2024 [40] to examine potential behavioural consequences of visual perturbations caused by *lens3*RNAi. Briefly, high-speed video recordings of hunting behaviour were obtained in horizontal and vertical arenas for 30-minute periods during which, in each trial, they were provided three *Aedes aegypti* larvae to hunt. Visual responses to prey items were scored blindly to estimate the following parameters: 1) Hunting success, i.e., percentage of prey caught and 2) Latency, which we define as the time it took larvae to capture the first prey item. Prior to recordings, larvae were starved for approximately 24 hours to facilitate responses to prey. A subset of trials were conducted under bright lighting conditions designed for larval vision as in Rathore et al., 2024 [40] (FigS4A,B in S1 File; white light with UV supplementation; horizontal arena = 6.27 x 1014 photons/cm2/s; vertical arena = 7.03 x 1014 photons/cm2/s). Another subset of trials were conducted in dim lighting conditions (FigS4C,D in S1 File; white light; horizontal arena = 9.29 x 1012 photons/cm2/s; vertical arena = 9.29 x 1012 photons/cm2/s) to challenge the visual system of larvae. Spectra of both arenas in dim and bright light environments were obtained using the Flame-S-XR1 spectrometer (Ocean Optics). To increase sample sizes for statistical tests, data from individuals injected with both RNAi probes were pooled for comparison against controls using Wilcoxon rank-sum tests.

## Supporting information

**S1 File. Supplementary files.**
(DOCX)

## Acknowledgments

We thank Yoshinori Tomoyasu for invaluable help and advice with RNAi. We sincerely thank Chet Closson and the Live Microscopy Core (UCMC, University of Cincinnati) for assistance with qPCR, Dr Melodie Fickenscher (CEAS, University of Cincinnati) for help with SEM imaging, Jessica Webster and the CCHMC for help with TEM imaging and members of the Buschbeck lab for helpful discussions.

## Author contributions

**Conceptualization:** Amartya T. Mitra, Shubham Rathore, Joshua B. Benoit, Elke Buschbeck.

**Data curation:** Amartya T. Mitra, Shubham Rathore, Elke Buschbeck.

**Formal analysis:** Amartya T. Mitra, Shubham Rathore, Augusta Jester, Ruby Hyland-Brown, Annette Stowasser.

**Funding acquisition:** Elke Buschbeck.

**Investigation:** Amartya T. Mitra, Shubham Rathore, Augusta Jester, J Hassert, Annette Stowasser.

**Methodology:** Amartya T. Mitra, Shubham Rathore, Elke Buschbeck.

**Project administration:** Elke Buschbeck.

**Resources:** Elke Buschbeck.

**Supervision:** Shubham Rathore, Elke Buschbeck, Annette Stowasser.

**Validation:** Amartya T. Mitra, Elke Buschbeck.

**Visualization:** Amartya T. Mitra.

**Writing – original draft:** Amartya T. Mitra, Shubham Rathore, Elke Buschbeck, Annette Stowasser.

**Writing – review & editing:** Amartya T. Mitra, Shubham Rathore, Elke Buschbeck, Annette Stowasser.

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
