## [Decision Letter · Decision Letter 0]

4 Mar 2025

PONE-D-25-03003Cataract induction in an arthropod reveals how lens crystallins contribute to the formation of biological glassPLOS ONE

Dear Dr. Buschbeck, Thank you for submitting your manuscript to PLOS ONE. After a thorough evaluation, we recognize its potential; however, it does not meet PLOS ONE's publication criteria due to several significant concerns. We encourage you to revise the manuscript, addressing the key issues highlighted during the review process. Please note that without substantial revisions, the chances of publication in this journal are limited.

We look forward to receiving your revised manuscript.

Kind regards,

Reza Yousefi, Ph.D

Academic Editor

PLOS ONE

“This study was supported by the National Science Foundation, USA under grant IOS-1856241.”

“We thank Yoshinori Tomoyasu for invaluable help and advice with RNAi. We sincerely thank Chet Closson and the Live Microscopy Core (UCMC, University of Cincinnati) for assistance with qPCR, Dr Melodie Fickenscher (CEAS, University of Cincinnati) for help with SEM imaging, Jessica Webster and the CCHMC for help with TEM imaging and members of the Buschbeck lab for helpful discussions. This study was supported by the National Science Foundation, USA under grant IOS-1856241.”

“This study was supported by the National Science Foundation, USA under grant IOS-1856241.”

4. Please update your submission to use the PLOS LaTeX template. The template and more information on our requirements for LaTeX submissions can be found at http://journals.plos.org/plosone/s/latex

Reviewers' comments:

Reviewer's Responses to Questions

**Comments to the Author**

1. Is the manuscript technically sound, and do the data support the conclusions?

Reviewer #1: Yes

Reviewer #2: Yes

2. Has the statistical analysis been performed appropriately and rigorously? 

Reviewer #1: Yes

Reviewer #2: Yes

3. Have the authors made all data underlying the findings in their manuscript fully available?

Reviewer #1: Yes

Reviewer #2: Yes

4. Is the manuscript presented in an intelligible fashion and written in standard English?

Reviewer #1: No

Reviewer #2: Yes

5. Review Comments to the Author

Reviewer #1: The article ‘Cataract induction in an arthropod reveals how lens crystallins contribute to the formation of “biological glass”’ (Amartya T. Mitra et al.) contains an interesting and original study of cataracts model in arthropods and can be recommended for publication with a few points.

1. The abstract is vaguely written and more data on the specific results obtained should be added.

2. The abbreviation FD is superfluous, it occurs only 2 times and can be done without it.

3. There is a lack of information on the protein composition of arthropod lenses, are there other proteins besides Lens3, this should be discussed and literature data should be cited.

4- Also missing are comments on the structural features of the Lens3 protein and its properties, if these are known, they should be given.

5. A clear and concise conclusion is missing.

Reviewer #2: 

This study by Mitra et al. explores the role of the Lens3 gene in lens development in Thermonectus marmoratus larvae, an arthropod species, using RNA interference (RNAi) to silence Lens3 and assess its effects on lens transparency, refractive properties, and visual function. The authors report that Lens3 knockdown induces cataract-like opacities, degrades image quality, and impairs hunting behavior under low-light conditions, while the eye's refractive state remains unchanged. This work is innovative and contributes valuable insights into lens biology and arthropod vision, though some issues require attention to enhance its scientific rigor and clarity.

Major Concerns:

• The manuscript speculates that Lens3 knockdown disrupts lens transparency via protein aggregates and chitin disorganization but lacks direct evidence or a clear mechanism. To clarify the mechanism, it would be beneficial to provide supporting data (e.g., biochemical assays) or explicitly note the current lack of evidence as a limitation for future research.

• The term "cataract" is used for lens opacities without sufficient quantitative or comparative justification. Strengthening the classification of lens opacities as "cataracts" by quantifying their size, density, and distribution, and comparing them to vertebrate cataracts, would enhance the analysis.

• Lens3 is not compared to lens proteins in other arthropods or vertebrates, limiting evolutionary insight. Adding a comparative analysis (e.g., sequence alignments or functional similarities) would help place Lens3 in a broader biological context.

• The unchanged refractive state despite image degradation is noted but underexplored. Expanding the discussion on why the refractive state remains unchanged despite image degradation, and exploring its implications for refractive development in arthropods versus vertebrates, would be beneficial.

Minor Concerns:

• The term "biological glass" is a key term in the title and text but is not defined, which may confuse readers. Adding a concise definition in the introduction would clarify its use in the context of lens biology.

• The use of "lens3RNAi" and "Lens3 knockdown" interchangeably may cause confusion. Using consistent terminology throughout the manuscript would improve clarity.

• In the cDNA preparation experiments, please clarify how you excluded the possibility of DNA contamination in the extracted RNA, as this could impact the accuracy of the qPCR results used to confirm Lens3 knockdown.

6. PLOS authors have the option to publish the peer review history of their article (what does this mean? ). If published, this will include your full peer review and any attached files.

**Do you want your identity to be public for this peer review?** For information about this choice, including consent withdrawal, please see our Privacy Policy .

Reviewer #1: No

Reviewer #2: No

---

## [Author Response · Author response to Decision Letter 1]

29 Apr 2025

Please see the response to review document, which includes comments regarding both reviewers as well as requested editorial changes.

---

## [Decision Letter · Decision Letter 1]

12 May 2025

Cataract induction in an arthropod reveals how lens crystallins contribute to the formation of biological glass

PONE-D-25-03003R1

Dear Dr. Elke Buschbeck,

We’re pleased to inform you that your manuscript has been judged scientifically suitable for publication and will be formally accepted for publication once it meets all outstanding technical requirements.

Kind regards,

Reza Yousefi, Ph.D

Academic Editor

PLOS ONE

Additional Editor Comments (optional):

Reviewers' comments:

Reviewer's Responses to Questions

**Comments to the Author**

1. If the authors have adequately addressed your comments raised in a previous round of review and you feel that this manuscript is now acceptable for publication, you may indicate that here to bypass the “Comments to the Author” section, enter your conflict of interest statement in the “Confidential to Editor” section, and submit your "Accept" recommendation.

Reviewer #2: All comments have been addressed

2. Is the manuscript technically sound, and do the data support the conclusions?

Reviewer #2: Yes

3. Has the statistical analysis been performed appropriately and rigorously? 

Reviewer #2: Yes

4. Have the authors made all data underlying the findings in their manuscript fully available?

Reviewer #2: Yes

5. Is the manuscript presented in an intelligible fashion and written in standard English?

Reviewer #2: Yes

6. Review Comments to the Author

Reviewer #2: Dear Authors,

The use of DNase treatment during RNA extraction is a critical step to minimize genomic DNA (gDNA) contamination in cDNA preparation. However, confirming the absence of residual gDNA via a no-reverse-transcriptase (-RT) control-using gDNA-specific primers-would strengthen the methodology. While agarose gel electrophoresis is insufficiently sensitive to detect trace gDNA, qPCR’s high sensitivity raises the possibility of amplification from even minimal contamination.

Nevertheless, gDNA contamination could lead to overestimation of expression levels. Furthermore, systematic contamination (if present) would likely affect all samples uniformly, including RNAi and control groups. Since the observed expression differences between groups remain consistent, the conclusions about gene expression trends remain valid. For future work, incorporating -RT controls would help definitively rule out gDNA-driven artifacts and enhance methodological rigor.

7. PLOS authors have the option to publish the peer review history of their article (what does this mean? ). If published, this will include your full peer review and any attached files.

**Do you want your identity to be public for this peer review?** For information about this choice, including consent withdrawal, please see our Privacy Policy .

Reviewer #2: No

---

## [Editor Report · Acceptance letter]

PONE-D-25-03003R1

PLOS ONE

Dear Dr. Buschbeck,

I'm pleased to inform you that your manuscript has been deemed suitable for publication in PLOS ONE. Congratulations! Your manuscript is now being handed over to our production team.

Kind regards,

on behalf of

Professor Reza Yousefi

Academic Editor

PLOS ONE